# Dating the origin and spread of specialization on human hosts in *Aedes aegypti* mosquitoes

Noah H Rose[1,2]*, Athanase Badolo[3], Massamba Sylla[4], Jewelna Akorli[5], Sampson Otoo[5], Andrea Gloria-Soria[6], Jeffrey R Powell[7], Bradley J White[8], Jacob E Crawford[8], Carolyn S McBride[1,2]*

[1]Department of Ecology and Evolutionary Biology, Princeton University, Princeton, United States; [2]Princeton Neuroscience Institute, Princeton University, Princeton, United States; [3]Laboratory of Fundamental and Applied Entomology, Université Joseph Ki-Zerbo, Ouagadougou, Burkina Faso; [4]Department of Livestock Sciences and Techniques, Sine Saloum University El Hadji Ibrahima NIASS, Kaffrine, Senegal; [5]Department of Parasitology, Noguchi Memorial Institute for Medical Research, University of Ghana, Accra, Ghana; [6]Department of Entomology. Center for Vector Biology & Zoonotic Diseases. The Connecticut Agricultural Experiment Station, New Haven, United States; [7]Yale University, New Haven, United States; [8]Verily Life Sciences, South San Francisco, United States

*For correspondence:
noahr@princeton.edu (NHR);
csm7@princeton.edu (CSMcB)

**Abstract** The globally invasive mosquito subspecies *Aedes aegypti aegypti* is an effective vector of human arboviruses, in part because it specializes in biting humans and breeding in human habitats. Recent work suggests that specialization first arose as an adaptation to long, hot dry seasons in the West African Sahel, where *Ae. aegypti* relies on human-stored water for breeding. Here, we use whole-genome cross-coalescent analysis to date the emergence of human-specialist populations and thus further probe the climate hypothesis. Importantly, we take advantage of the known migration of specialists out of Africa during the Atlantic Slave Trade to calibrate the coalescent clock and thus obtain a more precise estimate of the older evolutionary event than would otherwise be possible. We find that human-specialist mosquitoes diverged rapidly from ecological generalists approximately 5000 years ago, at the end of the African Humid Period—a time when the Sahara dried and water stored by humans became a uniquely stable, aquatic niche in the Sahel. We also use population genomic analyses to date a previously observed influx of human-specialist alleles into major West African cities. The characteristic length of tracts of human-specialist ancestry present on a generalist genetic background in Kumasi and Ouagadougou suggests the change in behavior occurred during rapid urbanization over the last 20–40 years. Taken together, we show that the timing and ecological context of two previously observed shifts towards human biting in *Ae. aegypti* differ; climate was likely the original driver, but urbanization has become increasingly important in recent decades.

## Editor's evaluation

This fundamental study by Rose and colleagues addresses key challenges in demographic inference in non-model systems with an innovative approach to model parameter calibration based on known historical events. Using this approach, they convincingly show that human specialization in *Ae. aegypti* mosquitoes likely evolved due to a past climate event around 5,000 years ago, and that recent rapid urbanization has continued to fuel its spread in West Africa in the past 20-40 years. This

work will be of broad interest to population geneticists working on demographic inference, and to mosquito biologists working on the monitoring and control of this important vector species.

## Introduction

The mosquito *Aedes aegypti* is thought to have originated as an ecological opportunist in forested areas of sub-Saharan Africa, and extant African populations of the subspecies *Ae. aegypti formosus* is relative generalists (*Figure 1A*). They breed in a wide variety of habitats and opportunistically bite a wide variety of human and non-human animals (*McBride et al., 2014*; *Rose et al., 2020*; *Xia et al., 2021*; *Diouf et al., 2021*). However, at some point in the past, a few populations in West Africa evolved to specialize in living with and biting humans, giving rise to a human-adapted subspecies *Ae. aegypti aegypti* that has since spread around the global tropics (*Figure 1A*) and become the world's most effective vector of the viruses responsible for yellow fever, dengue, Zika, and chikungunya (*McBride et al., 2014*; *Brady and Hay, 2020*; *Powell et al., 2018*; but see *Chadee et al., 1998* ).

Understanding the ecological factors that drive mosquitoes like *Ae. aegypti* to adapt to human hosts and habitats is important because it can help us predict and respond to future burdens of mosquito-borne disease on a planet whose climate and landscape are being transformed by human activity. Recent work suggests that human specialization in *Ae. aegypti* originally evolved as an adaptation to the extremely long, hot dry seasons of the Sahel region of West Africa (*Figure 1B*). *Aedes* mosquitoes lay their eggs just above the water line in small, contained bodies of water, and immature stages require water for development. The scarcity of natural tree-hole and rock-pool aquatic habitat for up to 9 months of the year in the Sahel likely forced *Ae. aegypti* to rely on human-stored water for survival, with adaptations to breeding in artificial containers (*Powell et al., 2018*; *Gillett, 1955*; *Petersen, 1977*; *Leahy et al., 1978*; *Metz et al., 2023*) and biting humans (*McBride et al., 2014*; *Rose et al., 2020*; *Gouck, 1972*) soon following. For example, adaptation to artificial containers involved changes in egg hatching behavior and larval starvation resistance (*Saul et al., 1980*) that allow immature stages to thrive in water that is relatively low in nutrients (e.g. bacteria and detritus) and high in oxygen. However, neither highly seasonal precipitation patterns nor human-derived pots of water were present in the Sahel until relatively recently in evolutionary time. Settled human societies practicing water storage in clay vessels likely developed within the last 10,000 years in sub-Saharan Africa (*Huysecom et al., 2004*; *Ehret, 2002*). Meanwhile, the Sahara Desert and associated Sahelian biome have only existed in their present form for the past 5000 years; the region was a vast savannah during the African Humid Period from 15,000–5000 years ago. If *Ae. aegypti* evolved to specialize in biting humans as a by-product of breeding among humans during the intense dry seasons of the West African Sahel (hereafter 'Sahel'), then the shift should have occurred no more than 5000 years ago.

The idea that *Ae. aegypti aegypti* first emerged when all elements of its current niche were finally present approximately 5000 years ago and has been suggested multiple times in the vector biology literature (e.g. *Powell et al., 2018*; *Petersen, 1977*; *Peter, 2016*), but has never been comprehensively tested with modern genomic data. A site-frequency-spectrum- (SFS-) based analysis of exome sequence data estimated that neighboring generalist and human-specialist populations in West Africa diverged about 15,000 years ago (*Crawford et al., 2017*), but this estimate was sensitive to model specification and parameterization. Conversely, a molecular clock-based phylogenetic analysis found that all African populations of *Ae. aegypti* have a common ancestor approximately 17,000–25,000 years ago, but did not precisely date the subsequent split between human-specialist and generalist subspecies (*Soghigian et al., 2020*).

In contrast to the uncertainty surrounding exactly when (and why) specialization occurred (*Figure 1C*), the historical forces that led to the spread of *Ae. aegypti aegypti* out of Africa are better understood. Human-specialist *Ae. aegypti* is thought to have first migrated to the Americas on ships during the Atlantic Slave Trade approximately 500 years ago (*Figure 1C*). Its arrival there coincided with the first recorded outbreaks of yellow fever in the New World in the 1600s (*Powell et al., 2018*). Subsequent outbreaks of yellow fever exacted an enormous toll in the following centuries, especially among immunologically naive European and Indigenous populations, shaping not just health outcomes, but also the political and economic history of the Americas (*McNeill, 2010*). Similarly, *Ae. aegypti*'s invasion of Asia and Oceania was followed by outbreaks of dengue, chikungunya, and later Zika, although yellow fever remains absent in these areas (*Brady and Hay, 2020*; *Powell et al., 2018*).

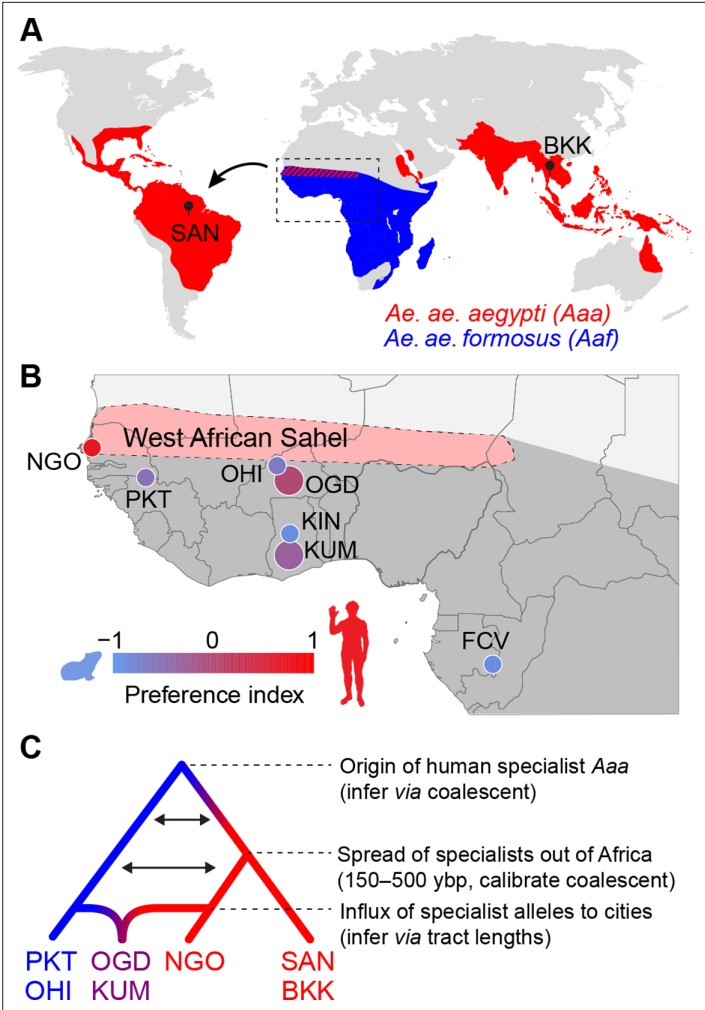

**Figure 1.** Dating the origin and spread of the human-specialist form of *Aedes aegypti*. (**A**) The human-specialist form of *Aedes aegypti aegypti* (*Aaa*) is thought to have originated in West Africa, before invading the Americas in association with the Atlantic Slave Trade and subsequently spreading across the global tropics. (**B**) Present-day *Ae. aegypti* populations in West Africa vary in their preference for human hosts. Large cities (e.g. KUM, OGD) are shown as larger circles, while small towns are shown as smaller circles. Map region corresponds to the inset in panel A. Pink shading marks the intensely seasonal Sahel region, where human specialists likely originated. Behavioral data taken from *Rose et al., 2020*. (**C**) The timing of major events in the evolutionary history of human-specialist populations (red lineages) can be inferred using population genomic approaches, such as coalescent analysis (for older events) and tract length analysis (for more recent events). SAN, Santarem, Brazil; BKK, Bangkok, Thailand; NGO, Ngoye, Senegal; PKT, PK10, Senegal; OHI, Ouahigouya, Burkina Faso; OGD, Ouagadougou, Burkina Faso; KUM, Kumasi, Ghana; KIN, Kintampo, Ghana; FCV, Franceville, Gabon.

Today, invasive populations of *Ae. aegypti* in the Americas and Asia are almost all closely related to each other and distinct from their ancestors in Africa (*Powell et al., 2018*; *Gloria-Soria et al., 2016*).

A third and putatively much more recent event in the evolutionary history of human-specialist *Ae. aegypti* involves a shift towards human biting in growing African cities that would otherwise be expected to harbor generalist populations. Regardless of local climate, urban *Ae. aegypti* mosquitoes in places like Kumasi, Ghana and Ouagadougou, Burkina Faso are consistently more responsive to humans than those from nearby rural areas (*Figure 1B*, compare KUM to KIN and OGD to OHI) (*Rose et al., 2020*). This shift is associated with the same underlying ancestry component that defines human specialists in the Sahel and outside Africa, suggesting that it results from an influx of human-specialist alleles into urban areas rather than independent evolution of human biting (*Figure 1C*; *Rose et al., 2020*). Although the exact timing of the observed influx in places like Kumasi and Ouagadougou

remains to be tested, it likely occurred recently; the cities involved have only reached the high levels of human population density associated with greater mosquito preference for humans within the past few decades (*United Nations, 2019*). Urban populations that lack substantial admixture from this human-specialist ancestry component have also been described and posited to represent a genetically independent adaptation to human environments (*Brown et al., 2011*). However, when their behavior has been tested, these populations do not show an increased preference for human hosts (*Rose et al., 2020*). This does not preclude other forms of adaptation to human habitats.

Here, we take advantage of existing population genomic data from diverse African and non-African populations (*Rose et al., 2020*) to infer the timing of two key events in the evolutionary history of the human-specialist form of *Ae. aegypti* (*Figure 1C*) and thereby more rigorously test key hypotheses concerning their underlying ecological drivers. First, we address the timing of the initial emergence of specialist populations in the Sahel *via* cross-coalescent analysis—a methodological framework developed for studying human evolution that has rarely been applied in other species. Importantly, we take advantage of the known migration of specialist *Ae. aegypti* out of Africa 150–500 years ago to calibrate the coalescent clock and thus obtain a much more precise estimate than would otherwise be possible. Second, we use ancestry tract length analysis to infer the timing of the recent shift towards human biting in large West African cities. We find that human specialists originated with the emergence of the modern Sahelian climate approximately 5000 years ago, while the influx of human-specialist ancestry into urban areas coincides with rapid urbanization over the past 20–40 years.

## Results
### Dating the origin of human specialists within *Ae. aegypti*

One way to date the evolution of specialization on humans in *Ae. aegypti* is to examine the time course of divergence between specialist and generalist populations using cross-coalescent analysis. Coalescent approaches model relationships between haplotypes at a locus going backward in time, using the steady accumulation of mutations to estimate when haplotypes merge into a single ancestral sequence (*Schiffels and Wang, 2020*). Using the Multiple Sequentially Markovian Coalescent (MSMC), one can characterize the distribution of times to coalescence across genome-wide loci (*Schiffels and Wang, 2020*). Patterns of relative cross-coalescence are particularly informative. This is the rate of coalescence of haplotypes from different populations relative to the rate of coalescence of haplotypes from the same population. It should start close to zero in the recent past when two populations are isolated, but eventually plateau at one when, going back in time, they have merged into a single ancestral population. Given a known mutation rate and generation time, patterns of relative cross-coalescence can be used to infer the timing of splits and subsequent gene flow between populations (*Schiffels and Wang, 2020*).

One hurdle associated with the effective application of cross-coalescent analysis to real-world genomic data is the need for phased genomes. To meet this challenge, we leveraged statistical approaches and recently published short-read sequence data for 389 unrelated *Ae. aegypti* individuals (*Rose et al., 2020*) to generate a large phasing panel for this species. First, we used *HAPCUT2* to pre-phase individuals (*Edge et al., 2017*). In this step, we used linkage information present in short reads and read pairs to locally phase adjacent variants; because nucleotide diversity in *Ae. aegypti* is high (about 2% *Rose et al., 2020*), multiple variants are often present within a single sequencing fragment, making pre-phasing highly effective in this species. Over 95% of prephase calls had maximal HAPCUT2 phred-scaled quality scores of 100, and prephase blocks (i.e. local haplotypes) were 728 bp long on average (interquartile range 199–1009 bp). We then used *SHAPEIT4.2* to assemble the prephase blocks into chromosome-level haplotypes, using statistical linkage patterns present across our panel of 389 individuals (*Delaneau et al., 2019*).

A second hurdle associated with coalescent analysis is the need for a reliable scaling factor that can be used to translate coalescent time into real-time. This parameter depends on the *de novo* mutation rate and generation time, neither of which is precisely known for *Ae. aegypti*. The mutation rate is likely on the order of $3 \times 10^{-9}$ per generation given estimates from other insects (*Drosophila melanogaster*: $2.8 \times 10^{-9}$, *Bombus terrestris*: $3.6 \times 10^{-9}$, *Heliconius melpomene*: $2.9 \times 10^{-9}$) (*Keightley et al., 2014*; *Keightley et al., 2015*; *Liu et al., 2017*), and most populations of *Ae. aegypti* probably go through ~15 generations per year (*Crawford et al., 2017*; *Gloria-Soria et al., 2016*; *Beserra*

*et al., 2006*; *Marinho et al., 2016*). However, there is substantial uncertainty in both numbers (10 or 12 generations per year are also common estimates, for example *Gloria-Soria et al., 2016*), and any inference of the timing of specialization derived from these values would be extremely rough.

To overcome this problem, we decided to leverage the known timing of the spread of human-specialist *Ae. aegypti* out of Africa to ground truth and calibrate our coalescent scaling factor. If the species did indeed escape Africa during the Atlantic Slave Trade, coalescent analysis of the relationship between human-specialist populations within and outside Africa should show a single strong pulse of migration that matches the timing of human-slave trafficking for plausible values of the mutation rate and generation time. If we then assume these values are constant through evolutionary time, we can use them in subsequent analyses of the older split between specialists and generalists.

We used *MSMC2* and *MSMC-IM* to fit an isolation-with-migration model of the relationship between a human-specialist population from the Sahel (Ngoye, Senegal [NGO], *Figure 1B*) and a human-specialist population from the invasive range (Santarem, Brazil [SAN], *Figure 1A*; *Schiffels and Wang, 2020*). As expected, this analysis revealed a single strong pulse of migration (*Figure 2A*, inset), which we presume corresponds to the Atlantic Slave Trade. NGO has experienced substantial admixture from nearby generalist populations and is almost certainly not the sole origin point of invasive populations. For example, populations in Luanda, Angola are more closely related to invasive populations in the Americas (*Kotsakiozi et al., 2018*) (see Discussion). For this reason, some coalescence events between NGO and SAN may correspond to the specialist-generalist split or other historical events, in addition to the invasion dynamics we are interested in. Nevertheless, NGO provides the best available proxy for human-specialist populations in their native habitat, and the observed migration signal should largely reflect the time course of their spread to the Americas.

We next obtained publicly available data on the number of enslaved people trafficked across the Atlantic in 25year intervals between 1500 and 1875 (*Liu et al., 2017*) and used the Bhattacharyya Coefficient to assess the degree of correspondence between this historical record and *MSMC*-inferred mosquito migration for given combinations of the mutation rate and generation time (*Figure 2A*). The Bhattacharyya Coefficient should take a value of zero in the case of non-overlapping distributions (red areas of the plot) and one in the case of perfect overlap (white areas). The fact that we see good overlap between the two distributions (yellow-white color) across a wide range of reasonable mutation rates and generation times for *Ae. aegypti* is consistent with our understanding of the species' recent history and supports our approach. For example, if we take the common literature value of 15 generations per year (0.067 years per generation) (*Crawford et al., 2017*; *Gloria-Soria et al., 2016*), the *de novo* mutation rate that maximizes correspondence between the two datasets is $4.85 \times 10^{-9}$ (black dot in *Figure 2A*, used in *Figure 2B*), which is on the order of values documented in other insects. We chose to carry forward this calibrated scaling factor (corresponding to any combination of values found along the line in *Figure 2A*) into subsequent analyses. Note that using Bangkok, Thailand (BKK, *Figure 1A*) as our reference invasive population instead of Santarem, Brazil produced a similar pattern, although the peak of the migration pulse was slightly older (*Figure 2—figure supplement 1M versus* N). This could be noise, since bootstrap replicates vary in the position of the highest peak (*Figure 2—figure supplement 1*). Alternatively, if BKK's progenitor population became isolated from South American populations relatively early on, it could be that it did not experience the subsequent migration that SAN experienced. In both cases, time-calibrated demographic histories showed a bottleneck in invasive populations about 500 years ago (*Figure 2—figure supplement 1A and B*).

We next sought to examine the historical relationship between human-specialist and generalist populations of *Ae. aegypti* within Africa using our newly calibrated scaling factor. We again used *MSMC2* and *MSMC-IM* to characterize patterns of cross-coalescence, but this time contrasting human specialists in Ngoye, Senegal (NGO, *Figure 1B*) with nearby generalists from PK10, Senegal (PKT, *Figure 1B*). We infer that these populations had similar effective population sizes ~10,000 years ago, which gradually expanded before diverging about 5000 years ago (*Figure 2C*). Isolation-with-migration analysis also suggested that these populations abruptly diverged about 5000 years before the present. This can be seen in the cumulative migration signal, which should reach one when (going back in time) populations have merged into a single ancestral population. Cumulative migration begins dropping slowly 15,000–30,000 years ago, suggesting a small amount of genetic structure in ancient times, but then drops precipitously 3000–5000 years ago (*Figure 2D*). This rapid divergence shows a striking correspondence to the end of the African Humid Period, as evidenced by the record

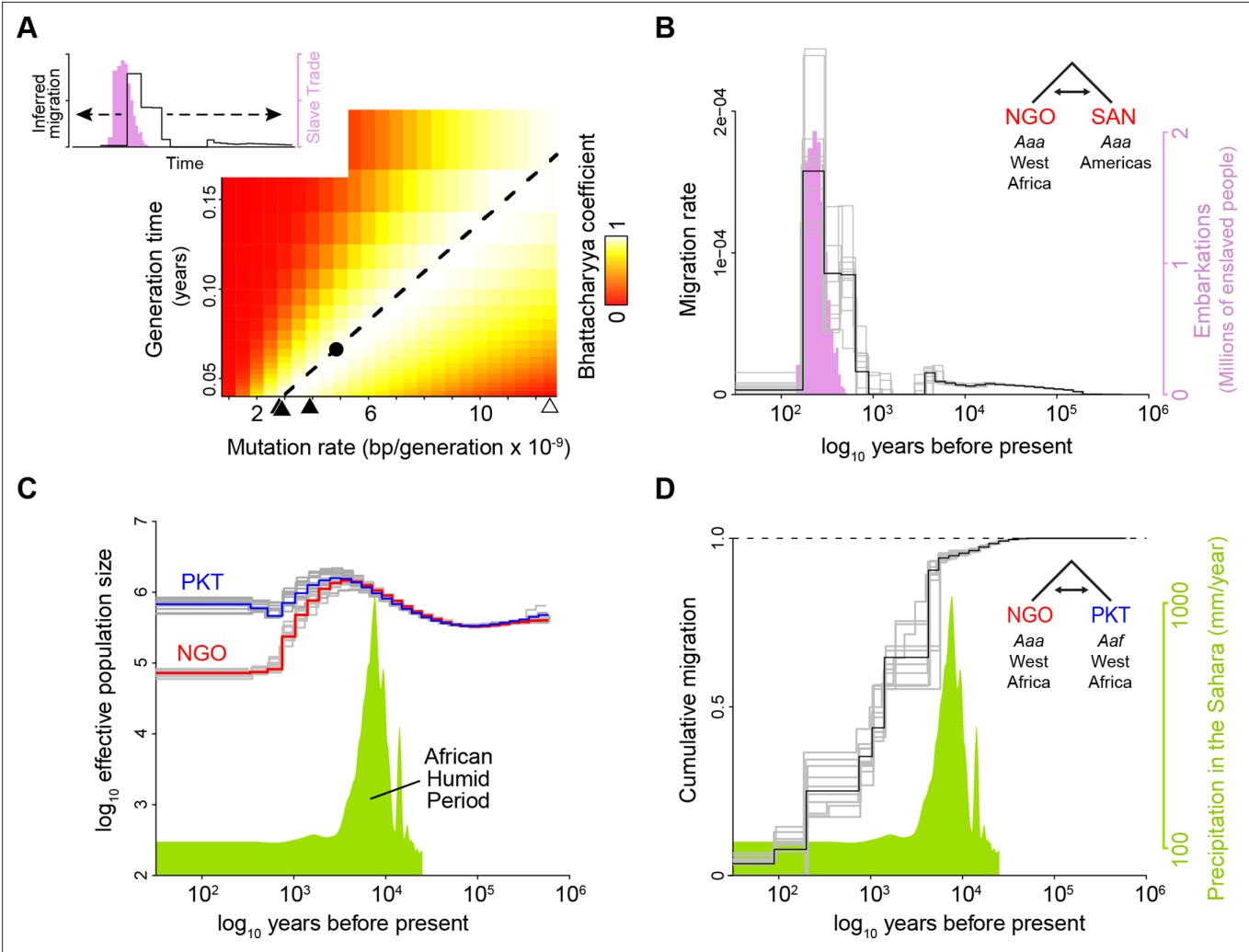

**Figure 2.** A calibrated coalescent scaling factor for *Ae. aegypti* suggests rapid evolution of human specialists at the end of the African Humid Period. (A–B) Calibration of a coalescent scaling factor for *Ae. aegypti*. (A), The Bhattacharyya coefficient reveals the extent of overlap between the timing of the Atlantic Slave Trade (based on historical records *Slave Voyages, 2023*, pink distribution in inset) and inferred migration of *Aaa* out of Africa (based on cross-coalescent analysis, black line in inset) for given combinations of the mutation rate (μ) and generation time (g). A value of one indicates perfect correspondence through time and 0 indicates no overlap. Estimated mutation rates for three other insects (see Results) and *Homo sapiens* are marked along x-axis with closed and open triangles, respectively. y-axis ranges from the shortest to longest reasonable generation times for *Ae. aegypti*. The dashed line indicates the μ/g ratio that provides the strongest match between genomic and historical data. The dot marks the exact calibration used in the other panels (g=0.067, μ=4.85 × 10⁻⁹), but any combination of μ and g that falls along the line would produce identical results. (B) The inferred timecourse of migration of *Aaa* from West Africa to the Americas shown alongside the historical record of the Atlantic Slave Trade that was used for calibration. When populations first split, the MSMC-IM model infers an instantaneous 'migration' of haplotypes to the new population. Older migration events may correspond to older coalescent events due to recent admixture in NGO. (C–D). Cross-coalescent analysis of the timing of human specialization.(C) Inferred effective population sizes for West African *Aaa* (NGO) and *Aaf* (PKT) superimposed on Saharan rainfall data (inferred from Atlantic marine sediments *Beserra et al., 2006*). (D) Estimated cumulative migration between *Aaa* (NGO) and *Aaf* (PKT) is expected to plateau at 1 going backwards in time, when populations have merged into a single ancestral population. Populations diverged rapidly at the end of the African Humid Period. In all panels, gray lines represent bootstrap replicates. This figure highlights analyses relevant to key questions (migration rate for the invasion process, cumulative migration for the original split of specialists and generalists). Full analyses for all population pairs are shown in *Figure 2—figure supplement 1*.

The online version of this article includes the following figure supplement(s) for figure 2:

**Figure supplement 1.** Full detail of cross-coalescent analyses comparing the Sahelian human specialists to invasive human specialists or nearby generalists.

of leaf wax isotopes present in marine sediments from the Atlantic Ocean floor (*Figure 2D*; *Tierney et al., 2017*). Analyses using an alternative generalist population (OHI, *Figure 1B*) yielded very similar results (*Figure 2—figure supplement 1*).

## Dating the origin of increased human specialization in urban populations

Within Africa, human-specialist populations of *Ae. aegypti* are largely restricted to the unusual Sahel climatic zone (*Figure 1A–B*). However, even in places where most mosquitoes are generalists, urban populations tend to be more responsive to human hosts than their rural counterparts (*Rose et al., 2020*). This increased willingness to bite humans is strongly correlated with levels of admixture from a shared human-specialist ancestry component found across locations (*Rose et al., 2020*), suggesting that the behavioral shift is the result of a recent influx of human-specialist alleles into rapidly growing cities. Before estimating the timing of this shift, we used the *f3* statistic to confirm that *Ae. aegypti* populations in the large cities of Kumasi, Ghana (KUM) and Ouagadougou, Burkina Faso (OGD), could be accurately modeled as mixtures of human-specialist and generalist populations (*Peter, 2016*). This statistic tests whether allele frequencies in a focal population are consistently intermediate between frequencies in two candidate source populations. If so, the genome-wide average *f3* should be negative, while a positive value provides evidence neither for nor against admixture. We used three different populations as candidate sources of human-specialist ancestry (West African NGO, American SAN, or Asian BKK, *Figure 1A and B*) and two different populations as candidate sources of generalist

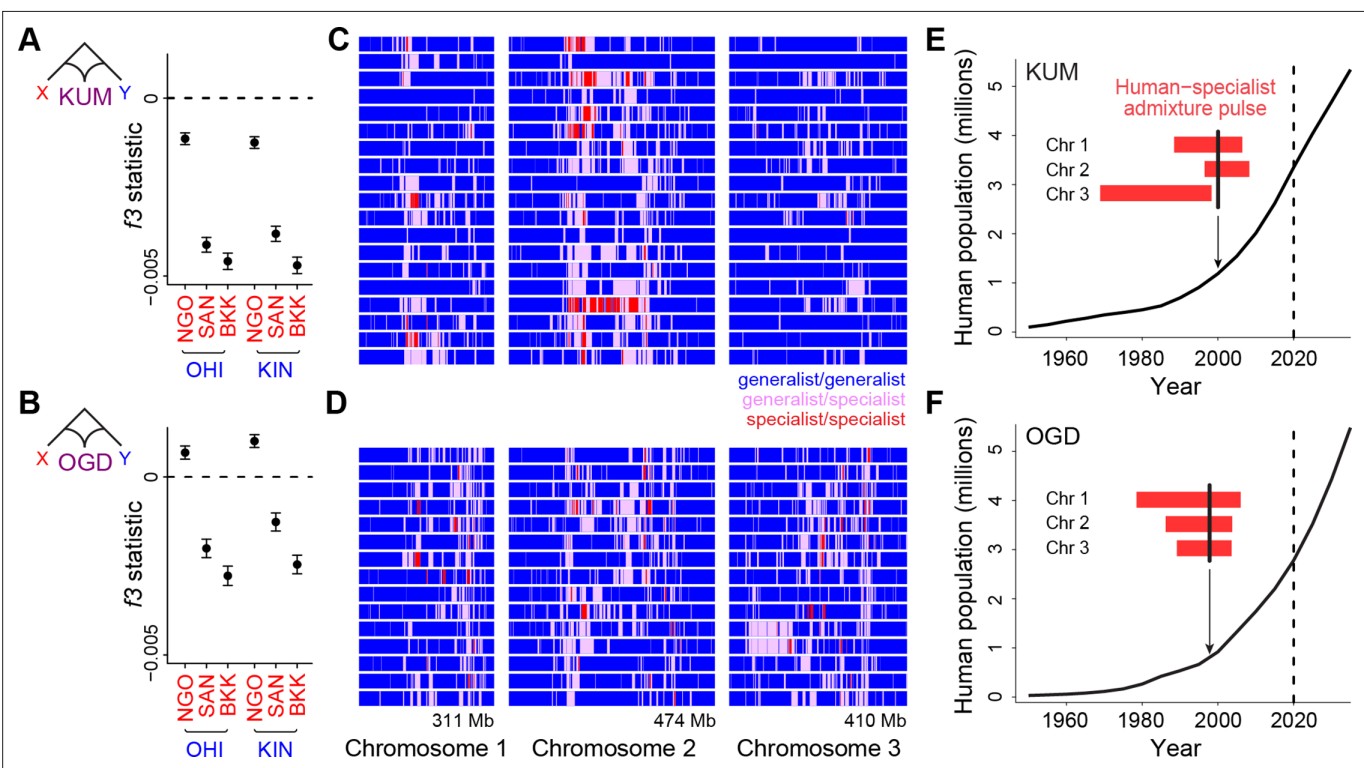

**Figure 3.** Long tracts of human-specialist ancestry in rapidly growing cities suggest a recent influx associated with modern urbanization. (A–B) *f3* analysis confirming that Kumasi, Ghana (A, KUM) and Ouagadougou, Burkina Faso (B, OGD) can be modeled as a product of admixture between generalist and human-specialist *Ae. aegypti* populations. Negative values provide evidence of admixture. Error bars show 95% jackknife confidence intervals. (C–D) Distribution of tracts of human-specialist ancestry (purple, heterozygous; red, homozygous) across 19 unrelated KUM genomes (C) and 15 unrelated OGD genomes (D). (E–F) The inferred timing of human-specialist admixture (red bars) overlaid on publicly available human population growth data and near-future growth projections for KUM (E) and OGD (F). Red bars correspond to 95% confidence intervals for each chromosome, and thick black lines mark the mean for all three chromosomes. Dashed lines mark the date when human population data were collected.

The online version of this article includes the following figure supplement(s) for figure 3:

**Figure supplement 1.** Simulations confirm the reliability of admixture analyses in West African cities.

**Figure supplement 2.** Tracts of human-specialist ancestry are concentrated in similar parts of the genome in Kumasi and Ouagadougou.

ancestry (West African KIN or OHI, *Figure 1B*). The resulting f3 values were substantially and significantly negative for both cities in almost all cases (*Figure 3A and B*). However, the use of NGO as a proxy for the human-specialist source resulted in only modestly negative values for KUM and positive values for OGD (*Figure 3A and B*). This ambiguous signal likely reflects the complex history of NGO, which is itself admixed despite its human-specialist ecology. The non-African specialists, in contrast, are geographically isolated from generalists. They may therefore be a better proxy for an unadmixed human-specialist source even if the specialist ancestry present in KUM and OGD came from within Africa. Denser sampling of human specialists both within and outside Africa will be necessary to come to a definitive conclusion about the true sources of admixture in these large cities.

To test whether the timing of admixture in KUM and OGD matches the timing of the rapid growth of human populations in each location, we turned to ancestry tract-length analysis. When human-specialist migrants first come into a generalist population, their offspring will have long tracts of human-specialist ancestry in their genomes (*Liang and Nielsen, 2014*). Over time, recombination should break these tracts into smaller and smaller pieces, making tract length an inverse correlate of the amount of time that has passed since the influx of human-specialist ancestry occurred (*Corbett-Detig and Nielsen, 2017*). These patterns can be used to investigate relatively recent gene flow, whereas coalescent approaches are informed by mutations that take thousands of generations to accumulate. We conducted ancestry tract analysis in KUM and OGD, as implemented in *AncestryHMM* (*Corbett-Detig and Nielsen, 2017*), using as source populations the human specialists and generalists that gave the strongest signals of admixture in our f3 tests: BKK and OHI.

We found long tracts of human-specialist ancestry in KUM and OGD, indicative of a relatively recent influx of human-specialist alleles (*Figure 3C and D*). Using the recombination map from the *Ae. aegypti* L5 assembly, and masking regions previously identified as under divergent selection between human specialists and generalists (see Methods) (*Rose et al., 2020*; *Matthews et al., 2018*), we

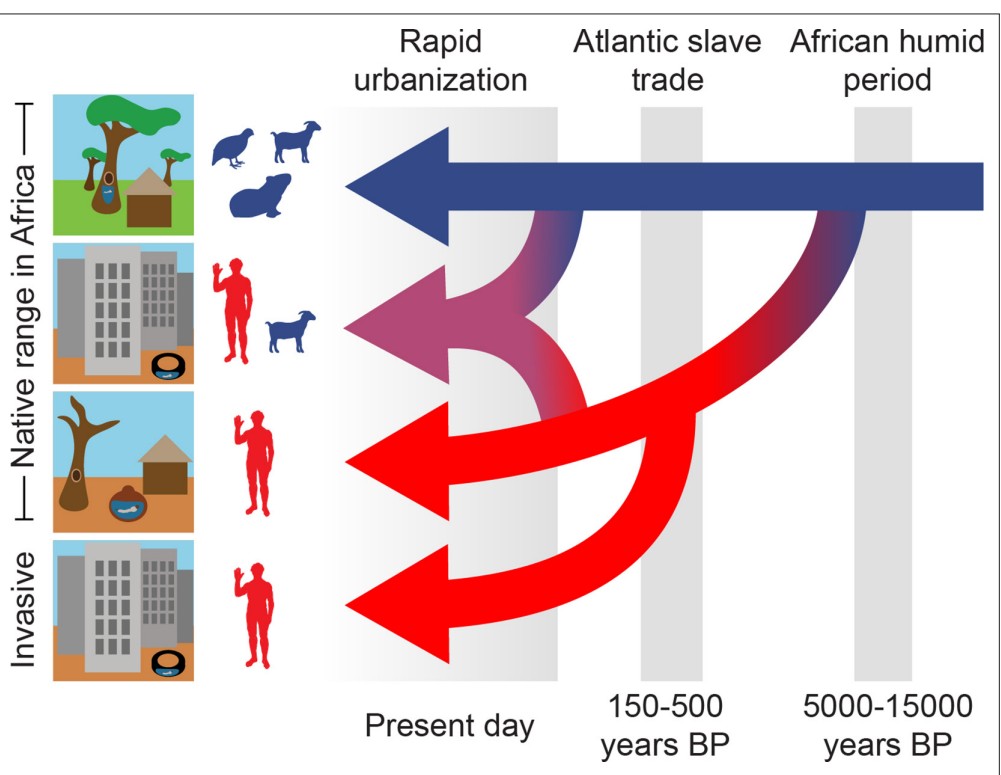

**Figure 4.** Three epochs define the origin and spread of the human-specialist form of *Ae. aegypti*. Our analyses suggest that human-specialized *Ae. aegypti* rapidly diverged from their generalist counterparts after the end of the African Humid Period, with the emergence of settled human societies in intensely seasonal habitats with long, hot dry seasons. Human specialists then migrated from West Africa to the Americas during the Atlantic Slave Trade. Finally, we find evidence of a recent influx of human-specialist ancestry into the rapidly growing cities consistent with an ongoing shift in the ecology of *Ae. aegypti* in present-day Africa.

estimated the number of generations of recombination since admixture. In both cities, admixture appears to have been recent, occurring within the last 20–40 years, coinciding with rapid human population growth in both cities (*Figure 3E and F*). The distribution of human-specialist tracts was not entirely uniform; instead, they were more concentrated in a few key regions (*Figure 3C and D*). This signal could not be explained by failure to detect admixture in specific regions as simulations confirmed that we had statistical power to identify mixing across the whole genome (*Figure 3—figure supplement 1*). Moreover, the presence of human-specialist ancestry was more spatially correlated across the genomes of individuals from the two cities than would be expected due to random chance (permutation p<0.001, *Figure 3—figure supplement 2*). This signal could reflect parallel patterns of selection for human-specialist ancestry in chromosomal regions that are important for urban adaptation. Overall, our results are consistent with a recent influx of human-specialist ancestry into the rapidly growing cities of Kumasi and Ouagadougou.

## Discussion

Taken together, our results suggest that the ecology of *Ae. aegypti* has been in flux across human history (*Figure 4*). We previously found that *Ae. aegypti* populations in their native range in sub-Saharan Africa were most specialized on human hosts in the West African Sahel, where they depend on human water storage for breeding across the long dry season (*Rose et al., 2020*). However, urban populations were also more attracted to human hosts than nearby rural populations regardless of climate (*Rose et al., 2020*). Here, we used cross-coalescent and ancestry tract analyses of genomic data to determine when these ecological shifts occurred and thus gain additional insight into their underlying drivers. According to our cross-coalescent estimates, human-specialist and generalist populations of *Ae. aegypti* diverged rapidly at the end of the African Humid Period, about 5000 years ago. In contrast, our ancestry tract analyses suggest that the urbanization effects are much more recent, playing out over the last 20–40 years.

The finding that human specialists emerged at the end of the African Humid Period is consistent with the hypothesis that climate provided the initial selective push toward specialization. It was at this time, approximately 5000 years ago, that the Sahara dried and the highly seasonal Sahel biome to its south came to be. *Ae. aegypti* mosquitoes inhabiting the area would have found themselves in a novel situation, devoid of rain-filled breeding sites for nine months of the year. This could have driven adaptation to human-derived breeding sites, such as the clay pots used for water storage by early pearl millet farmers (*Huysecom et al., 2004*; *Burgarella et al., 2018*). Interestingly, pearl millet domestication is also estimated to have occurred ~5000 years ago in the western Sahel (*Burgarella et al., 2018*), bringing multiple aspects of this mosquito's human-specialist niche together simultaneously in the region. In addition to the signal of rapid divergence 5000 years ago, we saw some evidence for steady population growth and accumulation of modest genetic structure starting 15,000–30,000 years ago. This observation is compatible with a model where some populations expanded into newly habitable parts of the present-day Sahara/Sahel at the start of the African Humid Period, giving rise to weak geographical structure, before rapidly diverging in response to a drying climate and the emergence of the human-specialist niche at its end. Although they seem to have occasionally interbred, specialist and generalist populations then became largely isolated and ecologically distinct.

It is widely accepted that human-specialist *Ae. aegypti* spread to the Americas (and eventually throughout the global tropics) during the Atlantic Slave Trade, ushering in a new era of vector-borne disease (*Powell et al., 2018*). We relied on this historical understanding to calibrate our coalescent scaling factor. However, our results also help establish its credibility. The cross-coalescent signal between Ngoye, Senegal, and Santarem, Brazil closely fit *a priori* expectations, even without calibration. Using literature values for average insect mutation rates and our best estimate of *Ae. aegypti* generation time would have also yielded good correspondence between the inferred timing of migration and the historical record of the Atlantic Slave Trade (*Figure 2A*). The duration and sudden drop-off in migration observed in the coalescent signal also closely match expectations from the historical record (*Figure 2A*).

Luanda, Angola was a major port during the Atlantic Slave Trade and likely served as the proximate source of many *Ae. aegypti* mosquitoes introduced to the Americas. *Ae. aegypti* populations there show substantial human-specialist ancestry, similar to those from Senegal (*Powell et al., 2018*; *Kotsakiozi et al., 2018*). This raises the question of whether human-specialist populations of *Ae.*

*aegypti* could have arisen in or around Angola rather than in the western Sahel. However, the closest generalist relatives of human-specialist populations are found in West Africa (*Kotsakiozi et al., 2018*). Present distributions of populations may not perfectly match their historical distributions, but the most parsimonious explanation is that human specialists (the 'proto-*Ae. aegypti*' described in *Powell et al., 2018*) first arose in the western Sahel and later spread to both Luanda and the Americas during the slave trade.

While our cross-coalescent analyses are remarkably consistent with prior expectations, several caveats and open questions remain. First, the temporal resolution of our inferences is relatively low, and both previously published simulations (*Wang et al., 2020*) and our own bootstrap replicates (*Figure 2B–D*, gray lines) suggest relatively wide bounds for the precise timing of events. Second, it is difficult for us to validate the long-range statistical phasing upon which the analyses are based. Overall linkage disequilibrium is relatively low in *Ae. aegypti*, dropping off quickly over a few kilobases and reaching half its maximum value within about 50 kb *Matthews et al., 2018*; this is likely sufficient for assembling shorter, high-confidence prephase blocks into longer haplotypes in many cases. However, phase-switch errors may be common across longer distances – potentially affecting inferences in the most recent time windows. Nevertheless, the similar results we obtain using different proxy populations (and thus different input haplotype structures) for human-specialist and generalist lineages (see *Figure 2—figure supplement 1*) suggests that our results are robust to potential mistakes in long-range haplotype phasing.

Another important source of uncertainty comes from our assumption that the coalescent scaling factor should maximize the overlap between the peak of estimated *Ae. aegypti* migration and the peak of the Atlantic Slave Trade (*Figure 2B*). If we instead consider alternative scenarios where peak migration occurred at the beginning of the slave trade era, around 1500, then our inferred mutation rate would be lower (about 2.4e-9, assuming 15 generations per year), pushing back the split of human-specialist lineages to ~10,000 years before present. This scenario seems less plausible, in part because our isolation-with-migration analyses suggest a gradual onset of migration between continents rather than a single, early-pulse model. It would also make it harder to explain the timing of the bottleneck observed in invasive populations; the first signs of this bottleneck occur at the beginning of the slave trade (~500 years ago) with our current calibration (*Figure 2—figure supplement 1A*), but would be pushed to a pre-trade date in this alternative scenario. We can also consider a scenario in which peak *Ae. aegypti* migration occurred more recently, perhaps around 1850, corresponding to increased global shipping traffic outside the slave trade alone. In this case, our inferred mutation rate would be higher (or generation time lower), and the split of human-specialist lineages would be placed at about 3000 years ago. Overall, the best match between the existing literature and our data corresponds to our main estimates, but alternative scenarios could gain support if future research finds evidence for a different time course of invasion than is suggested by the epidemiological record.

This study represents one of the first applications of cross-coalescent analysis to a non-model system. Cross-coalescent analyses are a powerful tool for understanding historical relationships between populations, but they are difficult to implement in practice, largely due to the logistical/computational burden of developing a large genome phasing panel and the difficulty of estimating an accurate coalescent scaling factor. Accurate calibration of coalescent analysis is notoriously difficult even in well-understood systems like humans (*Scally and Durbin, 2012*). Here, we take advantage of a continent-wide set of genomes, combined with read-based prephasing and population-wide statistical phasing, to develop a phasing panel that will enable future studies in *Ae. aegypti* with a lower barrier to entry. The same approach may work for other study organisms with similar population genomic properties; high levels of diversity are helpful for prephasing and at least moderate levels of linkage disequilibrium are important for the assembly of prephase blocks. Likewise, our use of the historical record to calibrate a coalescent scaling factor may be effective in other taxa whose history is characterized by known historical events as well as events with unknown timing. Similar calibrations have been used to understand the relationship between coalescent analyses and human history (*O'Fallon and Fehren-Schmitz, 2011*; *Schiffels and Durbin, 2014*), but the approach should be especially valuable in non-model organisms where mutation rates and generation times are less well characterized.

Human specialization first emerged within *Ae. aegypti* thousands of years ago, but now, in the face of the rapid urbanization in sub-Saharan Africa, generalist populations of this species seem to

be undergoing a new shift. We estimate an influx of human-specialist alleles into the rapidly growing cities of Kumasi and Ouagadougou within the past 20–40 years. Positive selection during adaptive introgression can increase tract lengths and make admixture appear more recent than it actually is. For this reason, we masked regions of the genome thought to underlie adaptation to human habitats before running our analysis. If selection has acted outside these regions, however, admixture may be somewhat older than we estimate. The exact geographic origin of the human-specialist ancestry tracts found in both cities also remains uncertain. They are likely derived from nearby specialist populations, with whom both KUM and OGD showed affinity in previous *ADMIXTURE* analyses (*Rose et al., 2020*). They could alternatively (or additionally) be derived from distant specialist populations or even invasive populations outside Africa; dormant *Ae. aegypti* eggs are often transported inadvertently, e.g., in the used-tire trade. More comprehensive sampling and analysis of human-specialist populations inside and outside Africa will be necessary to identify the precise origins of the genetic material currently transforming urban populations in Africa.

Regardless of the source, the influx of human-specialist ancestry into *Ae. aegypti* mosquitoes inhabiting African cities will likely have important implications for human health. Human-specialist *Ae. aegypti* are both more likely to bite humans and more likely to serve as competent vectors of dengue, Zika, and yellow fever (*Rose et al., 2020*; *Sylla et al., 2009*; *Aubry et al., 2020*; *Miller et al., 1989*). Ouagadougou has seen repeated outbreaks of dengue fever in recent years (*Im et al., 2020*; *Badolo et al., 2022*), and mosquitoes there are still genetically and phenotypically intermediate between generalists and specialists (*Rose et al., 2020*; *Kotsakiozi et al., 2018*; *Badolo et al., 2022*). It is possible that such populations will eventually become more specialized as urbanization continues. The emergence of urban yellow fever in Africa since 2016 may have also been hastened by the expansion of human-specialist populations of *Ae. aegypti* (*Kraemer et al., 2017*; *Klitting et al., 2018*). For these reasons, monitoring and control of *Ae. aegypti*, taking into account possible changes in the vector capacity of local populations over time, should remain a high priority in the native range of this key vector species.

## Methods

### Genome phasing

We developed a genomic phasing panel using 389 previously published *Ae. aegypti* genomes sequenced to 15 x coverage (*Rose et al., 2020*), including 359 from across the species' native range in Africa, as well as 30 from invasive populations outside of Africa (16 from Bangkok, Thailand, and 14 from Santarem, Brazil) (*Rose et al., 2020*). This panel also included four *A. mascarensis* individuals, but the phasing quality for these individuals may be lower due to the small number of individuals used. We used as input the full set of previously *bcftools*-called biallelic SNPs (*Rose et al., 2020*). We further filtered the panel to include variant sites with GQ >20 and DP >8, including only variants in putative single-copy regions (mean coverage 5–30 x), and excluding annotated centromeric and repeat regions, as well as the sex locus (*Matthews et al., 2018*). Ultimately, our phasing panel comprised 52,534,217 biallelic SNPs. We used a two-step phasing procedure. We first pre-phased nearby heterozygous sites using information present in sequencing reads within individuals using *HAPCUT2* (*Gloria-Soria et al., 2016*), and then carried out statistical phasing on a population level with *SHAPEIT4.2* (*Delaneau et al., 2019*), using a phase set error rate of 0.0001. Due to memory constraints, we carried out two separate *SHAPEIT4.2* runs – one with all samples from East and Central Africa, and a second with all samples from West Africa and outside of Africa. This strategy should maximize the use of phasing information from samples showing similar patterns of linkage disequilibrium (*Choi et al., 2018*).

### MSMC2 and MSMC-IM analyses

We used MSMC2 and MSMC-IM according to published best practices to carry out cross-coalescent and isolation-with-migration analyses, respectively (*Schiffels and Wang, 2020*; *Wang et al., 2020*). For this family of analyses, the inference may be less accurate if the recombination rate is much higher than the mutation rate. This constraint is not a problem for the human genomic data that these analyses were originally designed for but may present a barrier to analyses in other species (*Sellinger et al., 2021*). For *Ae. aegypti*, both recombination and mutation rates are likely about an order of magnitude lower than those observed in humans (*Crawford et al., 2017*; *Matthews et al., 2018*),

suggesting a similar ratio to that observed in humans (~1) and providing support for the application of these analyses to *Ae. aegypti*. We first generated genome-specific masks for each genome using the script bamCaller.py, which is provided with the MSMC2 package. We also used a general genomic mask that included only putative single-copy regions (mean coverage 5–30 x) and excluded annotated centromeric and repetitive regions. This general genomic mask further excluded regions previously identified as under selection between specialists and generalists (*Rose et al., 2020*), as well as the sex locus, and sites that were annotated as uncallable using SNPable on the African consensus of the AaegL5 assembly, with parameters -l 150 r 0.5 (*Li, 2023*). We used bcftools to extract phased genomes for focal individuals from our phasing panel. We then carried out cross-coalescent analyses using MSMC2. We carried forward the resulting MSMC2 output into MSMC-IM analysis, fitting isolation-with-migration models to the fitted cross-coalescent estimates, with default regularization settings. We bootstrapped MSMC2 and MSMC-IM analyses using ten replicates of three 400 Mb chromosomes composed of resampled blocks of 20 Mb.

We characterized rates of cross-coalescence and historically effective population sizes among five populations. We used Ngoye, Senegal (NGO) as a representative of human specialists living in Africa. We used Ouahigouya, Burkina Faso (OHI), and PK10, Senegal (PKT, a forest area outside of Kedougou, Senegal) as representatives of nearby generalist populations in West Africa. We used Bangkok, Thailand (BKK) and Santarem, Brazil (SAN) as representatives of invasive human-specialist populations outside of Africa. For each population, we used two representative individuals, one male, and one female. We confirmed the sex of these individuals by checking rates of read mapping at the sex locus (chromosome 1, positions 151680000–152950000) — males all had >700 counts per million, and females all had <400 counts per million. Note, however, that the sex locus was masked for coalescent analyses.

As discussed in the Results, our cross-coalescent analyses yielded results in coalescent units that need to be scaled by mutation rate and generation time to yield dates for key events. These parameters are difficult to calibrate even in well-characterized systems like humans and *Drosophila*. For this reason, we used the well-described relationship between the Atlantic slave trade and the spread of *Ae. aegypti* out of Africa to ground truth and calibrate our coalescent scaling factor. We used the Bhattacharyya coefficient (implemented in a custom R function) to measure the degree of concordance between the inferred *Ae. aegypti* migration rate between West Africa and the Americas (using Ngoye, Senegal, and Santarem, Brazil as reference populations) and the intensity of the Atlantic slave trade as documented in the Slave Voyages database (*Slave Voyages, 2023*). We then used the coalescent scaling factor that gave the strongest concordance between inferred migration and slave trade intensity (corresponding to a mutation rate of $4.85 \times 10^{-9}$, assuming 15 generations per year) to calibrate our subsequent analysis of divergence between human-specialist and generalist lineages.

The climate history of the Sahara and Sahel regions has been reconstructed from analysis of plant-derived marine sediments from the Eastern Atlantic (*Tierney et al., 2017*). We visualized the African Humid Period by using *ImageJ/FIJI* (*Rasband, 2015*) to trace the four reconstructed records of precipitation over the past 25,000 years (GC27, GC37, GC49, GC69) in the Sahel and Sahara described by *Tierney et al., 2017*. We averaged these traces, interpolating levels of precipitation with a 100 year step using the *R* function *approxfun* (*R Development Core Team, 2013*), and plotted the rolling average of this trace with a 1000 year window and 100 year step.

## f3 analyses

In order to determine how best to model potentially admixed urban populations in KUM and OGD, we used *f3* analyses of our previously called set of genome-wide SNPs with minor allele frequency >0.05 (*Rose et al., 2020*), as implemented in the program *threepop* (*Pickrell and Pritchard, 2012*), to test for signatures of admixture between different human-specialist (BKK, SAN, NGO) and generalist (OHI, KIN) populations. We used block-jackknifed Z-scores to assess whether the resulting values were significantly negative (*Peter, 2016*).

## AncestryHMM analyses

We used AncestryHMM to detect tracts of human-specialist ancestry in KUM and OGD, using a single-pulse model, running a separate analysis for each chromosome. We filtered the previously described biallelic SNP set (*Rose et al., 2020*) to include only ancestry-informative sites characterized by an

allele-frequency difference between populations of >0.3 and a minimum of 10 sampled alleles per population. Sites were then thinned such that no two were closer than 10,000 bp apart. We estimated recombination rates between sites using the recombination map published with the AaegL5 genome assembly (*Matthews et al., 2018*). For each analysis, we used reference panels of 10 unrelated and unadmixed OHI individuals and 16 unrelated and unadmixed BKK individuals (*Rose et al., 2020*). *AncestryHMM* requires estimates of admixture proportions for each chromosome as part of its input. We estimated admixture proportions on each chromosome using the set of 1,000,000 unlinked variants used to estimate genome-wide admixture proportions in a previous *ADMIXTURE* analysis (*Rose et al., 2020*; *Alexander et al., 2009*), subsetting these variants for each chromosome. *AncestryHMM* simultaneously estimates admixture timing during model training – we used 80 bootstrap resampling blocks of 1000 ancestry-informative sites to construct 95% confidence intervals for admixture pulse timing for each chromosome in each population. Since selection associated with adaptation to urban habitats could shape lengths of admixture tracts, we masked regions previously identified as under selection between human specialists and generalists when estimating admixture timing—namely, the outlier regions in *Rose et al., 2020*. However, we used an unmasked analysis to determine and visualize the genome-wide distribution of ancestries (*Figure 3*).

## Comparison of admixture patterns between cities

In order to determine whether the distribution of human-specialist ancestry across the genome was more similar between OGD and KUM than would be expected from random chance, we used circular chromosome permutation tests (*Stainton et al., 2015*). We first used Viterbi posteriors from each individual within a single population to calculate a population-specific local ancestry proportion at each site used in our *AncestryHMM* analysis and then calculated the Pearson correlation of local ancestry in KUM with that in OGD across all genomic loci. In each permutation, we randomly selected a new starting point for each chromosome, placing the preceding sequence at the end of the permuted chromosome but otherwise leaving the spatial relationship between loci undisturbed. We then recorded the Pearson correlation coefficient between the permuted distributions of ancestry in KUM and OGD, constructing a null distribution across 1000 permutations.

## AncestryHMM simulations

In order to be sure that the distribution of admixture tracts found by our *AncestryHMM* analysis reflects actual differences in local ancestry and not differences in our power to detect them, we carried out a series of simulations taking advantage of our phasing panel. We simulated heterozygous tracts of different sizes (500 kb, 1 Mb, 2 Mb, 10 Mb) derived from the first haplotype of a phased BKK genome (sample Debug_010_aegypti_Bangkok_Thailand_01) on the background of two different unadmixed OHI individuals (Debug023_aegypti_Ouahigouya_BurkinaFaso_F_11, Debug023_aegypti_Ouahigouya_BurkinaFaso_M_10), using *bedtools random* to generate random intervals in which to place the tracts, *bcftools* to extract the tracts from the BKK genome, and *bedtools intersect* to generate simulated input files for *AncestryHMM*. For each of the four tract lengths, we carried out 100 replicate simulations. We noticed that *AncestryHMM* found tracts of specialist ancestry across all simulations using a given OHI individual in some small regions of the genome. However, these regions differed between the two OHI individuals used in our simulations. We, therefore, suspect these regions are small *bona fide* tracts of human-specialist ancestry in these otherwise unadmixed OHI individuals rather than general features of the analysis or reference panels used.

## Acknowledgements

This manuscript is dedicated to the memory of Gilbert Bianquinche. This research was supported by the New York Stem Cell Foundation and a postdoctoral fellowship to NHR. from the Helen Hay Whitney Foundation. CSM is a New York Stem Cell Foundation – Robertson Investigator.

## Additional information

### Competing interests

Bradley J White, Jacob E Crawford: is affiliated with Verily Life Sciences. The author has no financial interests to declare. The other authors declare that no competing interests exist.

## Funding

| Funder | Grant reference number | Author |
|---|---|---|
| Helen Hay Whitney Foundation | | Noah H Rose |
| New York Stem Cell Foundation | | Carolyn S McBride |

The funders had no role in study design, data collection and interpretation, or the decision to submit the work for publication.

## Author contributions

Noah H Rose, Conceptualization, Data curation, Formal analysis, Investigation, Visualization, Methodology, Writing – original draft, Writing – review and editing; Athanase Badolo, Massamba Sylla, Jewelna Akorli, Sampson Otoo, Conceptualization, Investigation, Writing – review and editing; Andrea Gloria-Soria, Jeffrey R Powell, Bradley J White, Jacob E Crawford, Conceptualization, Data curation, Investigation, Writing – review and editing; Carolyn S McBride, Conceptualization, Supervision, Investigation, Visualization, Methodology, Writing – original draft, Writing – review and editing

## Author ORCIDs

Noah H Rose http://orcid.org/0000-0001-7129-4753
Athanase Badolo http://orcid.org/0000-0002-6652-4240
Jewelna Akorli http://orcid.org/0000-0002-3972-0860
Andrea Gloria-Soria http://orcid.org/0000-0002-5401-3988
Carolyn S McBride http://orcid.org/0000-0002-8898-1768

## Decision letter and Author response

Decision letter https://doi.org/10.7554/eLife.83524.sa1
Author response https://doi.org/10.7554/eLife.83524.sa2

# Additional files

## Supplementary files

• MDAR checklist

## Data availability

Scripts and processed data are available at https://github.com/noahrose/aaeg-evol-hist (copy archived at swh:1:rev:3d3b779b241244f8f17d873f316e7baedb930e18). Raw genomic data are available in the NCBI SRA at accession PRJNA602495. Phasing reference panel is available at https://doi.org/10.5061/dryad.2bvq83btk.

The following dataset was generated:

| Author(s) | Year | Dataset title | Dataset URL | Database and Identifier |
|---|---|---|---|---|
| Rose NH | 2022 | Dating the origin and spread of specialization on human hosts in *Aedes aegypti* mosquitoes | https://doi.org/10.5061/dryad.2bvq83btk | Dryad Digital Repository, 10.5061/dryad.2bvq83btk |

The following previously published dataset was used:

| Author(s) | Year | Dataset title | Dataset URL | Database and Identifier |
|---|---|---|---|---|
| Rose, et al | 2020 | Climate and Human Population Density Drive Adaptation to Human Hosts in *Aedes aegypti* Mosquitoes Across Africa | https://www.ncbi.nlm.nih.gov/sra/PRJNA602495 | NCBI Sequence Read Archive, PRJNA602495 |

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
