## [Editor Report]

This fundamental study by Rose and colleagues addresses key challenges in demographic inference in non-model systems with an innovative approach to model parameter calibration based on known historical events. Using this approach, they convincingly show that human specialization in *Ae. aegypti* mosquitoes likely evolved due to a past climate event around 5,000 years ago, and that recent rapid urbanization has continued to fuel its spread in West Africa in the past 20-40 years. This work will be of broad interest to population geneticists working on demographic inference, and to mosquito biologists working on the monitoring and control of this important vector species.

---

## [Decision Letter]

**Decision letter after peer review:**

Thank you for submitting your article "Dating the origin and spread of specialization on human hosts in *Aedes aegypti* mosquitoes" for consideration by *eLife*. Your article has been reviewed by 1 peer reviewer, and the evaluation has been overseen by a Reviewing Editor and George Perry as the Senior Editor. The following individual involved in the review of your submission has agreed to reveal their identity: Nicolas Lou (Reviewer #1).

The Reviewing Editor has drafted this to help you prepare a revised submission.

Before getting to the details, I first wanted to apologize for the exceedingly long amount of time that it took to reach this decision, and offer some explanation. We had three reviewers accept the invitation to review, and received one review promptly, but not the other two. Since there were two outstanding reviews I continually assumed that at least one would be submitted, if late… and with allowance for the intervening holidays and busy end-of-year period (and of course just the generally challenging times being faced by many), this stretched until now. To be brief from here, I am choosing to proceed with the positive decision on your paper at this point with one formally-submitted review, rather than add further time to the process to solicit wholly new reviews. This is partly due to my own comfort in evaluating your methods, results, and conclusions, as an unofficial second reviewer in this case. I fully support the comments of the submitted review and will add that I really like the paper – the methodological approach and the biological results are each outstanding on their own; together this is an outstanding paper in my view, with only relatively small changes needed prior to acceptance and publication.

Essential revisions:

Expand discussion regarding the specific assumptions made in your analyses, sources of uncertainty regarding the inputs, and how this impacts the certainty of your results. The inclusion of a formal uncertainty analysis could be warranted, with the expansion of this discussion to a mini-section of the paper.

---

## [Author Response]

Essential revisions:Expand discussion regarding the specific assumptions made in your analyses, sources of uncertainty regarding the inputs, and how this impacts the certainty of your results. The inclusion of a formal uncertainty analysis could be warranted, with the expansion of this discussion to a mini-section of the paper.

We have included more details about the phasing process, linkage structure, and our level of certainty about phasing and resulting inferences. We have also expanded our discussion of our assumption that the recent migration signal should correspond to the peak of the Atlantic Slave Trade, and discussed how deviations from this assumption would affect our conclusions.